# Biodegradation of Petroleum Hydrocarbons by *Drechslera*
*spicifera* Isolated from Contaminated Soil in Riyadh, Saudi Arabia

**DOI:** 10.3390/molecules27196450

**Published:** 2022-09-30

**Authors:** Rasha M. Al-Zahrani, Fatimah Al-Otibi, Najat Marraiki, Raedah I. Alharbi, Horiah A. Aldehaish

**Affiliations:** Department of Botany and Microbiology, College of Science, King Saud University, P.O. Box 22452, Riyadh 11495, Saudi Arabia

**Keywords:** *Drechslera spicifera*, biosurfactants, hydrocarbons, biodegradation

## Abstract

Currently, the bioremediation of petroleum hydrocarbons employs microbial biosurfactants because of their public acceptability, biological safety, and low cost. These organisms can degrade or detoxify organic-contaminated areas, such as marine ecosystems. The current study aimed to test the oil-biodegradation ability of the fungus *Drechslera spicifera*, which was isolated from contaminated soil samples in Riyadh, Saudi Arabia. We used hydrocarbon tolerance, scanning electron microscopy, DCPIP, drop-collapse, emulsification activity, recovery of biosurfactants, and germination assays to assess the biodegradation characteristics of the *D. spicifera* against kerosene, crude, diesel, used, and mixed oils. The results of DCPIP show that the highest oxidation (0.736 a.u.) was induced by crude oil on the 15th day. In contrast, kerosene and used oil had the highest measurements in emulsification activity and drop-collapse assays, respectively. Meanwhile, crude and used oils produced the highest amounts of biosurfactants through acid precipitation and solvent extraction assays. Furthermore, the biosurfactants stimulated the germination of tomato seeds by more than 50% compared to the control. These findings highlight the biodegradation ability of *D. spicifera*, which has been proven in the use of petroleum oils as the sole source of carbon. That might encourage further research to demonstrate its application in the cleaning of large, contaminated areas.

## 1. Introduction

Petroleum hydrocarbons are the most essential raw source of energy material for a myriad of industries [1]. Petroleum hydrocarbon pollutants are recalcitrant compounds, ranked as priority pollutants [2]. The previous, engineering-based physicochemical techniques for ex situ treatment of contaminants involved the usage of costly and dangerous materials, such as potassium permanganate and/or hydrogen peroxide as chemical oxidants to mineralize petroleum; threatening the lives of many organisms in contaminated areas [3]. In order to resolve the issue of increased costs and limited efficiency of conventional physicochemical methods, alternative technologies for in situ applications have developed, specifically, techniques based on the biological remediation capacity of plants and microorganisms [1,3].

Green technologies that function through biological means have been used in the bioremediation of petroleum pollutants and are economically efficient, versatile, and environmentally friendly [3]. Bioremediation involves the use of microorganisms to clean pollutants by degrading or detoxifying organic contaminated areas. Due to its safety, public acceptability, and low cost, it has become one of the most rapidly growing areas in environmental microbiology [4]. Bioremediation of sites polluted by crude oil is often restricted by poor biodiversity of indigenous microflora and/or scarcity of native microbes that possess the complementary substrates that are specifically required to degrade the different hydrocarbons found at polluted sites [3].

Efforts to biodegrade oil products involved both bacteria and fungi. These microorganisms possess sufficient metabolic capability to utilize petroleum carbon for cell synthesis, unlike any other biological species [5]. Specifically, bacteria are considered primary degraders and the most active agents in petroleum pollutant degradation [2,6]. Fungi are more successful degraders than bacteria for some hydrocarbons, such as polyethylene [7]. Their physiology and adaptation to hydrocarbon toxicity variations in the environment give them higher tolerance [4]. Fungi secrete extracellular enzymes that assimilate complex carbohydrates, ramify rapidly, and then digest on the substratum [8]. Their large-scale production can be achieved under severe environmental conditions, such as stress [9]. Furthermore, the filamentous structure of fungi allows their easy separation through filtration [8]. Fungi can potentially eliminate a wide range of environmental pollutants; similarly, multiple studies have reported their degradation ability towards polycyclic aromatic compounds, phenols, halogenated phenolic compounds, polychlorinated biphenyls, and petroleum hydrocarbons [6].

In different ecosystems, specific microorganisms have been reported to degrade different groups of molecules. Certain types degrade aliphatic hydrocarbons, mono- or poly-aromatics, where others degrade resin [10,11]. Bacterial hydrocarbon degraders include species of genera *Achromobacter*, *Acinetobacter*, *Arthrobacter*, *Azoarcus*, *Brevibacterium*, *Cellulomonas*, *Corynebacterium*, *Flavobacterium*, *Marinobacter*, *Micrococcus*, *Nocardia*, *Ochrobactrum*, *Pseudomonas*, *Stenotrophomaonas* and *Vibrio* [3,6,7,8,9,10,11,12]. Fungi degraders include those of genera *Aspergillus*, *Amorphoteca*, *Fusarium*, *Graphium*, *Neosartoria*, *Paecilomyces*, *Penicillium*, *Sporobolomyces*, and *Talaromyces*, and yeast of genera *Candida*, *Pichia*, *Pseudozyma*, *Rhodotorula*, and *Yarrowia* [13,14,15].

As Saudi Arabia is one of the largest producers of petroleum, several studies have been conducted to screen the existence of different oil biodegrades. Previous studies have shown that the strains *Lichtheimia ramose*, *Aspergillus polyporicola*, *Aspergillus spelaeus*, and *Aspergillus niger*, isolated from soil samples surrounding crude oil reservoirs in Saudi Arabia, may be considered significant oil-biodegrades [5,15].

In the current study, and for the first time, we evaluated the bio-degradation capabilities of fungus *D. spicifera*, isolated from contaminated soil samples in crude oil reservoirs in Riyadh, Saudi Arabia. The main goal of the current study was to isolate and recognize *D. spicifera* communities and examine their ability to employ different hydrocarbons as a distinct carbon source, proving its biodegradation capabilities. Furthermore, we sought to determine the effect of the isolated biosurfactants on the germination of tomato seeds to evaluate their safe application in different ecosystems.

## 2. Results

### 2.1. Identification of D. spicifera Strains in Different Soil Samples

In the current study, *D. spicifera* was isolated from contaminated crude oil reservoirs in Al Faisaliyyah and Ghubairah districts, Riyadh, Saudi Arabia. The identification of the isolated strain was confirmed microscopically in the Department of Botany and Microbiology, College of Science, King Saud University, Riyadh, Saudi Arabia. The incidences of isolated fungi in PDA cultures were 26.67% and 13.33% in the soil samples from Al Faisaliyyah and Ghubairah, respectively. All cultures demonstrated the capability of *D. spicifera* to grow with crude oil as the sole hydrocarbon, something which was demonstrated by the formation of a clear halo around each colony (data not shown).

### 2.2. Corresponding Reactivity between D. spicifera and Different Hydrocarbons

The ability of *D. spicifera* to tolerate the effect of different types of petroleum hydrocarbons was tested. The fungus was cultured in solid mineral salt medium (MSM) media supplemented with 1, 5, and 10% of different hydrocarbons (Table 1, Figure 1). The tested hydrocarbons showed variable decreases in the diameter of *D. spicifera* growth area, which created clear halos surrounded the colonies (Figure 1A–F). The calculated fungal dose inhibition response (DIR) to the tested hydrocarbons was significantly reduced in the cultures treated with crude oil (70.1%, *p* = 0.003), kerosene (62.6%, *p* < 0.001), diesel (56.1%, *p* < 0.001), and mixed oil (50%, *p* < 0.001) (Figure 1G).

The hydrocarbon tolerance was tested in liquid MSM medium (Figure 2). As shown, the growth rate of *D. spicifera* differed depending on the carbon source used. This was evidenced by the change of color in each flask (Figure 2A–F). The increase in weight was measured daily for a month and compared with the control. The highest growth weight was observed with kerosene, followed by crude oil, mixed oil, diesel, and used oil, respectively (Figure 2G). At the end of experiment, the dry weight of different cultures was calculated (Figure 2H). When compared with the control, all hydrocarbons resulted in higher growth of *D. spicifera.* Used oil (5.59 g), diesel (4.03 g), crude oil (3.42 g), and kerosene (2.24 g) showed significant increases in the dry weight of *D. spicifera* after 30 days of incubation (*p* < 0.001).

Furthermore, the SEM images of *D. spicifera* showed morphological changes in the structure of the isolates treated with 1% crude oil (Figure 3). Although *D. spicifera* showed an ability to grow on 1% crude oil, this caused some deformities in its cellular growth. The healthy fungi had a lean and homogeneous appearance with a wrinkled surface and intact structure (Figure 3A,B). The treated fungi appeared shredded into small twisted pieces, with inconspicuous conidia spore morphology (Figure 3C,D).

### 2.3. D. spicifera Biosurfactants Induced Hydrocarbons Bioremediation

In the current study, the oxidizing effects of *D. spicifera* caused a decolorization of the DCPIP redox dye, indicating the biodegradation of different hydrocarbons (Appendix A). The colorimetric analysis of treated flasks of *D. spicifera* was screened every three days until 15 days (Figure 4A). Comparison with the untreated control showed an increase of the DCPIP color absorbance at 450 nm in all tested hydrocarbons (Table 2). The highest oxidation was induced by crude oil at the 15th day (0.736 a.u.), followed by used oil (0.514 a.u.), where the rest of the tested hydrocarbons had similar absorbance at almost 0.4 a.u.

Otherwise, the emulsification activity of *D. spicifera* was tested against different treatments (Appendix A). The results show variable mixing of tested oils with cell free supernatant (CFS), indicating an increase in their emulsification. The calculated percentages (Figure 4B) show an increase in oil emulsification activities to almost 50% with CFS, as compared with water.

On the other hand, the drop-collapse assay showed a remarkable increase in the collapsing effect of *D. spicifera* bio-surfactants against tested hydrocarbons (Table 3). The results reveal that crude oil, diesel, and kerosene had the weakest collapsing effects as compared with the positive control of Triton X-100. Mixed and used oils had moderate collapsing effects induced by the surfactants produced by *D. spicifera*.

The above findings indicate that *D. spicifera* bio-surfactants changed the physical characteristics of the tested hydrocarbons.

### 2.4. Assessment of the Weight of Bio-Surfactants Recovered from D. spicifera

The current study highlighted the relationship between the amount of the bio-surfactants produced by *D. spicifera*, incubated with different carbon sources. It achieved this by using different assays. The acid precipitation assay showed that the amount of the produced precipitate differed according to the type of hydrocarbon (Figure 5A). The dry weight of bio-surfactants in the settings of crude oil was the highest (4.3 g), followed by used oil (3.78 g), where the lowest production was for mixed oil (1.52 g). Similarly, the solvent extraction assay showed that crude oil was the highest (5.25 g), followed by used oil (4.66 g), where the lowest production was for mixed oil (1.53 g).

Two other methods, the ammonium sulfate and zinc sulfate precipitation assays, were also used. In contrast with other assays, crude and used oils had the lowest amounts of bio-surfactants, where kerosene had the highest production.

### 2.5. Bio-Surfactants of D. spicifera Improved the Growth of Tomato Seeds

The germination of 100 *Solanum lycopersicum* (tomato) seeds was observed in the presence of *D. spicifera* bio-surfactant for 20 days to investigate the toxicity level (Figure 6A). Against expectations, the addition of bio-surfactants to tomato seeds did not inhibit their growth, but increased it. This might indicate the higher environmental safety levels of *D. spicifera* bio-surfactants. The highest significant growth was noticed at 12- and 15-days post-treatment with germination of 51% and 62%, respectively, as compared with the control (*p* < 0.01). This study has shown that stimulated tomato seeds with *D. spicifera* biosurfactants had germinated better than in control settings. Such biopreparation treatments particularly improve the plant germination on a contaminated base.

## 3. Discussion

*D. spicifera* or *Bipolaris spicifera* is the taxonomical nomenclature of a fast-growing seed-borne fungus. Morphologically, it is characterized by dark-brown colonies, sympodial conidiogenesis in the geniculated conidiophores, the conidial transverse distosepta, absence of the protuberant hilum, and bipolar germination [16,17]. In the current study, the *D. spicifera* strain was isolated from contaminated soil samples. The identification of the isolated strain relied on the morphological characteristics of the Conidia and Conidiophores, which confirmed its taxonomy [18]. This identification method has been used in previous studies from northern Sumatra and Italy as the only identification method [19,20]. In agreement with our findings, a previous study that was conducted in Dammam city, Saudi Arabia, showed that *D. spicifera* was identified in 7.8% of soil samples collected from oil tanker stations of the Aramco company [1]. Similarly, the soil samples collected from contaminated oil stations in the Suez Gulf in Egypt, indicated the existence of *D. spicifera* in the water bottom and sediment of the samples [21]. Another study has indicated the identification of *D. spicifera* in sub-aerial, confined, non-hypogean environments, such as deteriorated indoor wall paintings, another source of hydrocarbons, in Italy [22].

In the current study, the hydrocarbon tolerance of *D. spicifera* was tested in both solid and liquid media. The results indicate the formation of clear zones surrounding the colonies of *D. spicifera*. In contrast, no clearing zones were formed in control samples without any source of hydrocarbons, which indicate that the clearance zones resulted from fungal action and not because of other abiotic factors. Furthermore, our results indicate that the highest reduction in fungal growth was achieved with crude oil, while the lowest effect was achieved by the used oil treatment. The clearance zones surrounding colonies might be correlated with the presence of extracellular lipase enzyme [23]. A previous study has been conducted on petroleum hydrocarbons contaminated soil samples from India, and identified the presence of *Drechslera halodes*, which causes hydrocarbon enzymatic bioremediation by 37 ± 0.7 U/mL of extracellular lipase [24]. Similar findings indicated the formation of emulsified holes in coated plates with crude oil after incubation for 7–10 days with *Bacillus subtilis* and *Pseudomonas* sp. [25]. Another study concluded that the culture supernatant of *Trichosporon asahii* (yeast), when inoculated with diesel oil, produced a clear zone [26]. A study conducted by Ting et al., (2009) reported that *Pseudomonas lundensis* produced clearing zones when incubated with paraffin, mineral, and crude oils by metabolizing the long-chain-length of alkanes [27]. Based on the mean findings, the presence of clearing zones might be considered evidence of oil biodegradation resulting from the fungal growth and extracellular metabolic products.

In the current study, the fungal dry biomass of the culture broth after biodegradation was also measured. The highest amount of dry biomass was recorded by *D. spicifera* in used oil (5.585 g/L) followed by the lowest biomass in mixed oil (0.975 g/L). *D. spicifera* allowed biodegradation of different hydrocarbons, which oxidized DCPIP redox dye 15 days post-treatment. Furthermore, mixing *D. spicifera* with different hydrocarbons changes their physical characteristics, which was indicated by the increase in their emulsification by up to 50% and an increase in the collapsing effects. Previous studies have showed that the fungi of *Drechslera* sp., *Fusarium* sp., and *Papulaspora* sp. tolerated crude-oil biodegradation in the desert salt marsh in Kuwait, despite a high salt concentration and temperature [28,29,30]. Another study has shown that *D. spicifera* isolated from three oil stations in Egypt could degrade solid anthracene (28.6%) and the water-soluble fractions of crude oil (66.21%) after 14 days [21]. Most fungi that utilize petroleum hydrocarbons as a source of carbon and energy metabolized the molecules to CO_2_ and biomass [31]. Another study showed a correlated higher surface activity of biosurfactant solution with the larger diameter produced by *Lactococcus lactis* [32]. This indicates the ability of *D. spicifera* to grow in severe environmental conditions, in which any carbon source can be used for its survival.

The most significant characteristic of a potential hydrocarbon degrader is the ability to produce microbial biosurfactants, which compromise a mixture of organic (proteins, fatty acids, exopolysaccharides, and amino acids) and inorganic components [33]. The ability of these ingredients to emulsify the hydrocarbon through reduction of the interstitial surface tension tolerates the oil-bioremediation and oil recovery [34]. In the current study, *D. spicifera* increased the emulsification of the different hydrocarbons used. The weight of surfactants produced by *D. spicifera* was estimated and showed that the highest amount was produced by crude and used oils. Several studies have shown the ability of some fungal species to produce degrading enzymes, such as catalases, peroxidases, and laccases, which increase the degradation and immobilization of organic and inorganic contaminants [35,36]. Previous studies have shown the ability of members of the *Drechslera* genera to degrade aromatic hydrocarbons [24,37].

The effects of microbial biosurfactants on the biodegradation of different hydrocarbons from different countries have been well studied, however, almost no studies have been conducted about *D. spicifera* in Saudi Arabia. In a study from India, *Pseudomonas aeruginosa* showed an efficient production of biosurfactant with crude oil, which degraded ~82% of the petroleum hydrocarbons after 35 days [12]. Another Indian study has revealed the ability of biosurfactants produced by *Bacillus subtilis* (4.85 g/L) to degrade 97% of crude oil samples, which was explained by the synthesis of the bio-degradative enzymes of alkane hydroxylase and alcohol dehydrogenase [38]. The biosurfactants produced by *Trichoderma atroviride*, *Aspergillus nidulans*, and *Aspergillus sydowii* isolated from contaminated soil samples showed an efficient degradation (66.9–83.5%) of crude oil after 30 days [39]. Another study from Saudi Arabia used a mixture of biosurfactants produced by *Alternaria alternata*, *Aspergillus flavus*, *Aspergillus terreus*, and *Trichoderma harzianum,* which caused successful biodegradation (73.6%) of crude oil [40]. All of these studies, as well as our findings, suggest the robust oil-biodegradative ability of fungal biosurfactants, which might be explained by the production of degradative enzymes that can decrease the hydrocarbon pollution in contaminated spots.

Finally, to test the toxic effects of biosurfactants produced by *D. spicifera*, their germination activity was explored using tomato seeds for twenty days. Our results show a significant increase in the seed’s germination, which reflects their higher environmental safety levels. In agreement with our findings, a previous study did not report any toxicity against the growth of seeds of *Brassica oleracea* (Wild cabbage), *Solanum gilo* (Scarlet eggplant), and *Lactuca sativa* (Lettuce) treated with biosurfactants of *Candida lipolytica* [41]. Another study showed that the biosurfactants produced by *Pseudomonas* sp. stimulate the germinations of maize, lupine, pea, mustard, and oat seeds [42].

## 4. Materials and Methods

### 4.1. Samples Collection and Fungus Isolation

Six soil samples were collected from polluted crude oil reservoirs in two main districts, Al Faisaliyyah (two samples) and Ghubairah (four samples), in Riyadh city, Saudi Arabia. The samples were collected from the contaminated soils at a depth of 10 cm in sterile glass flasks and directly stored at 4 °C. The fine particles (<2.5 mm) were sieved by a Fieldmaster Soil Sampling Sieve Set (Cole-Parmer Inc., Vernon Hills, IL, USA).

To identify and sperate *D. spicifera* strain, an amount of one gram of each sample were premixed with 100 µL of Tween 80 and cultured into Sabouraud dextrose agar (SDA) plates for three days at 28 ± 2 °C [43]. The strains were identified microscopically, then the growing colonies were isolated and re-plated on a potato dextrose agar (PDA) plate as described in a previous study [44]. The morphological characteristics and taxonomy of *D. spicifera* were confirmed, microscopically, according to characteristics of the segmentation of the spores and conidia/Conidiophores structures [16] using the guidelines of the pictorial atlas of soil and seed fungi of Watanabe [18].

### 4.2. Hydrocarbons Biodegradation

Fresh oil hydrocarbons were obtained from Aramco Co. (Dammam, Saudi Arabia) in sterile bottles. These samples included crude oil, kerosene, diesel, and used oil. Furthermore, equal volumes of these hydrocarbons were mixed together to form the “mixed oil”. All samples sterilized by filtration and the biodegradation ability of *D. spicifera* were tested with a mineral salt medium (MSM), as described in previous studies [1,45].

### 4.3. Hydrocarbon Tolerance Test

The ability of the fungal strain to use different hydrocarbons as a sole source of carbon was tested by testing the growth rate on solid and liquid medium. The growth of *D. spicifera* on solid medium of MSM agar was little modified from previous studies [46,47]. Briefly, 0.5 mm of *D. spicifera* was inoculated on solidified MSM agar plates that were previously supplemented with 1%, 5%, or 10% of either crude oil, used oil, diesel, kerosene, or mixed oil. The hydrocarbon tolerance capacity was estimated by calculating the mycelial growth and DIR for ten days post-inoculation. DIR was calculated as follows:(1)$$\mathrm{DIR}=\frac{\mathrm{Growth}\;\mathrm{rate}\;\mathrm{on}\;\mathrm{hydrocarbon}\;\mathrm{plate}}{\mathrm{Growth}\;\mathrm{rate}\;\mathrm{on}\;\mathrm{control}\;\mathrm{plate}}\times 100$$

The growth of *D. spicifera* on liquid medium of MSM agar has been described in a previous study [48]. An amount of 200 mL of MSM in a sterile flask was mixed with 1% of any of tested hydrocarbons as a sole carbon source. Two slices (0.25 cm^2^) of fungus growing on solid MSM from a previous step were added to each flask, then incubated at 25 °C for 30 days while shaking (140 rpm). The growth rate and weight of fungus in grams were measured every three days.

Furthermore, the fungal dry biomass of the culture broth, post-biodegradation, was measured at the end of the 30-day incubation period [49]. The biomass was filtered by a grade 4 Whatman filter paper (Merck & Co., Inc., Kenilworth, NJ, USA), washed with chloroform, then dried at 60 °C for 24 h. After cooling, the amount of dried biomass was measured by a high precision analytical scale (Thermo Fisher Scientific, Waltham, MS, USA). All experiments were performed in triplicate.

### 4.4. Scanning Electron Microscopy (SEM)

The morphological characteristics of *D. spicifera* were investigated by SEM according to the methodology described by Al-Otibi et al. (2021) [50]. The microscopy work was undertaken in the Prince Naif Research Centre, King Saud University, Riyadh, Saudi Arabia. The normal morphology of isolated fungus was compared with another specimen pretreated with 1% of crude oil. The slides were prepared and tested by the JEOL JEM-2100 microscope (JEOL, Peabody, MA, USA), according to the manufacturer instructions.

### 4.5. DCPIP Assay

Crude oil degradation was tested by testing the effect of hydrocarbons oxidation induced by *D. spicifera* on the electron acceptor, DCPIP [51,52]. Briefly, MSM culture media was mixed with 1% of one of the tested hydrocarbons in addition to 0.1% Tween 80 and DCPIP (0.6 mg/mL). A piece of fungal hyphae of *D. spicifera* (1 cm^2^) was added to each hydrocarbon-specific mixture and incubated on a shaker at 25 °C for two weeks. Then, the colorimetric changes in DCPIP color were analyzed spectrophotometrically, at 420 nm, by BioTek Synergy™ 2 plate reader (Agilent Technologies, Inc., Santa Clara, CA, USA).

### 4.6. Drop-Collapse Test

The biosurfactants produced by *D. spicifera*, that induced oil-biodegradation, were assessed by screening the ability of the CFS drops to spread or collapse over an oil-coated solid surface [53]. CFS was collected from the *D. spicifera* cultured on MSM liquid medium for 30 days by centrifugation at 10,000 rpm (4 °C) for 30 min. CFS was filtered with a 0.45 μm membrane, then 10 µL was added over the surface of glass slides that contained a drop of 100 µL of crude oil. The drop collapse was estimated at 10× power of inverted microscope and results were interrupted as shown by [54]. A drop diameter of at least 0.5 mm was considered positive (+), where the drops larger than those produced by distilled water and by culture medium was considered negative.

### 4.7. Emulsification Activity Measurement

In the current study, the ability of biosurfactants produced by *D. spicifera* to increase the oil emulsification was tested [55]. Briefly, a tube of 2 mL of each oil was mixed with the same volume of CFS, prepared as described above. Another tube was mixed with water to act as a negative control. The tubes were homogenized at high speed and incubated overnight at room temperature. The emulsification activity was calculated as follows:(2)$$\mathrm{Emulsification}\;\mathrm{activity}\;\left(\%\right)=\frac{\mathrm{Height}\;\mathrm{of}\;\mathrm{emulsion}\;\mathrm{layer}}{\mathrm{Total}\;\mathrm{height}}\times 100$$

### 4.8. Recovery of Biosurfactants

Different techniques were used to isolate and estimate the amount of biosurfactants produced by *D. spicifera* CFS’s.

In the acid precipitation method [56], 3 mL of each CFS was adjusted by 6N HCl to a strong acidic pH of 2. After overnight incubation at 4 °C. A mixture of chloroform and methanol (2:1) was added to each tube (*v*/*v*) and re-incubated overnight at room temperature. A light-brown-colored precipitate was formed and collected by spinning down at 10,000 rpm, 4 °C for 30 min. The precipitate was dried and weighed.

In the solvent extraction method [57], CFS was mixed with an equal volume mixture of methanol, chloroform, and acetone (1:1:1). The mixture was homogenized for five hours at 200 rpm (30 °C). A white-colored precipitate was formed at the tubes bottom, which was isolated, dried, and weighed.

A further two precipitation methods by ammonium sulfate and zinc sulphate [58] were applied. CFS’s were incubated with either 40% (*w*/*v*) of ammonium sulfate or zinc sulfate overnight at 4 °C. Then, the mixture was centrifuged at 10,000 rpm for 30 min at 4 °C. In the CFS–zinc sulfate mixture, a light-brown precipitate was formed, isolated, dried and weighed. Using the CFS–zinc sulfate method, but for the CFS–ammonium sulfate mixture, the precipitate was extracted by acetone, isolated and dried under a fume hood. The resulting white-cream-colored powder was weighed. All experiments were compared with a negative control in which the CFS’s were replaced with distilled water.

### 4.9. Assessment of Toxicity Using the Germination Assay

The toxicity levels of isolated biosurfactants were assessed by testing their effect on the germination of Solanum lycopersicum (tomato) seeds against water (as a control) [59,60]. Briefly, 100 seeds of tomato were pre-treated with sodium hypochlorite, washed three times with distilled water, then soaked overnight in 50 mL of CFS with shaking. The treated seeds were washed and planted in a Petri dish containing a Whatman filter paper, incubated in the dark, and sprayed with 10 mL of water twice a day for 20 days. The growth rate was recorded in terms of the number of healthy growing plants. The percentage of germination was calculated as follows:(3)$$\mathrm{Seed}\;\mathrm{germination}\;\left(\%\right)=\frac{G\;\left(t\right)}{G\;\left(c\right)}\times 100$$
where G (t) is the number of seeds germinated CFS and G (c) is the number of seeds germinated in the control.

### 4.10. Statistical Analysis

All experiments were performed in triplicate. *One-way ANOVA* and *Dunnett’s* tests were used to estimate the significance levels at *p* < 0.05. Statistical analyses were performed using the SPSS statistical package (version 22).

## 5. Conclusions

In conclusion, the current study evidenced the biodegradation of different hydrocarbons by the biosurfactants produced by *D. spicifera*. The fungus changes the physical properties of the tested hydrocarbons by increasing their emulsification or solubility in water, which might prove the fungal efficacy in clearing the oil contamination from polluted sites. Furthermore, the produced fungal biosurfactants developed the germination of tomato seeds, which proved their safe usage as oil bioremediates. Different factors might modulate this process, which may urge great research interest in examining the possible pathways and strategies to develop cheap and safe bioremediation techniques for petroleum hydrocarbon polluted environments.

## Figures and Tables

**Figure 1 molecules-27-06450-f001:**
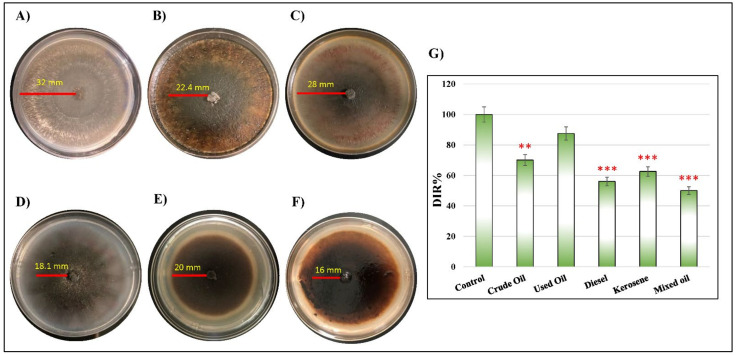
The ability of *D. spicifera* to tolerate effects of different hydrocarbons in solid medium. The fungal growth was tested in solid medium of MSM supplemented with 1% of different hydrocarbons. (**A**) MSM media without any treatments (control), (**B**) crude oil, (**C**) used oil, (**D**) diesel, (**E**) kerosene, (**F**) mixed oil, and (**G**) DIR percentages of different treatments. ** indicates *p* < 0.01 and *** indicates *p* < 0.001.

**Figure 2 molecules-27-06450-f002:**
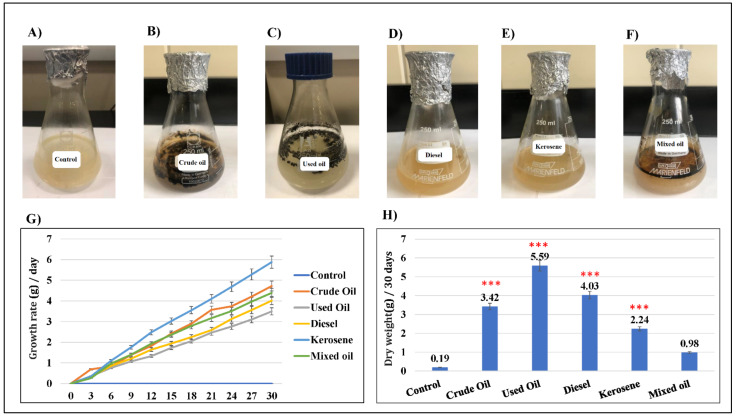
The ability of *D. spicifera* to tolerate effects of different hydrocarbons in liquid medium. The growth of *D. spicifera* was tested in liquid MSM media for 30 days in the presence of 1% of different hydrocarbons. (**A**) MSM media without any treatments (control), (**B**) crude oil, (**C**) used oil, (**D**) diesel, (**E**) kerosene, (**F**) mixed oil, (**G**) real-time measurement of the fungal growth in grams with different treatments, (**H**) comparison of the *D. spicifera* dry weight in grams after 30 days. *** indicates *p* < 0.001.

**Figure 3 molecules-27-06450-f003:**
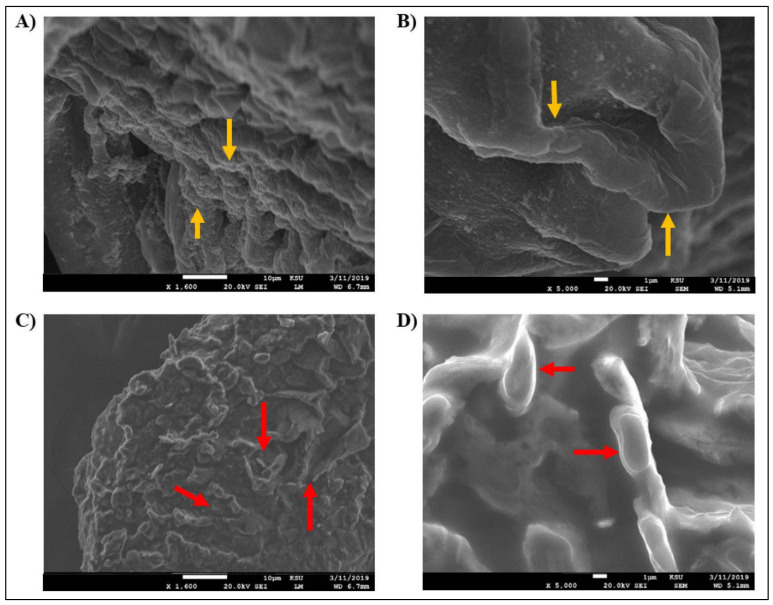
SEM imaging results of *D. spicifera*. The fungal growth was visualized and captured with JEOL JEM-2100 microscope. (**A**) At 1600× and (**B**) 5000×, images show the untreated samples as a control, with the yellow arrows indicating the lean and homogeneous fungal structures with wrinkled and intact structure. (**C**) At 1600× and (**D**) 5000×, images show the fungus treated with 1% crude oil, the red arrows show the fungus fragments as they were cut into small and twisted pieces, where in (**D**) the arrows show the inconspicuous conidia spore morphology.

**Figure 4 molecules-27-06450-f004:**
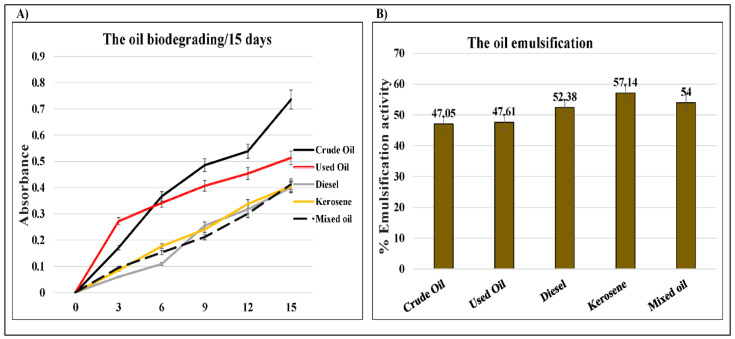
Biodegradation ability of *D. spicifera.* The fungal growth effects on the physical characteristics of tested hydrocarbons were evaluated with 1% of each oil. (**A**) DCPIP assay for 15 days, (**B**) percentage of emulsification activity.

**Figure 5 molecules-27-06450-f005:**
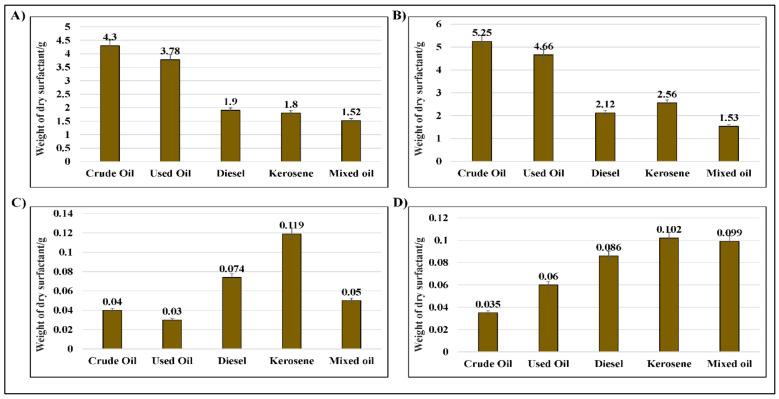
Bio-surfactant recovery assays of *D. spicifera*. (**A**) Acid precipitation, (**B**) solvent extraction, (**C**) ammonium sulfate, and (**D**) zinc sulfate precipitation assays.

**Figure 6 molecules-27-06450-f006:**
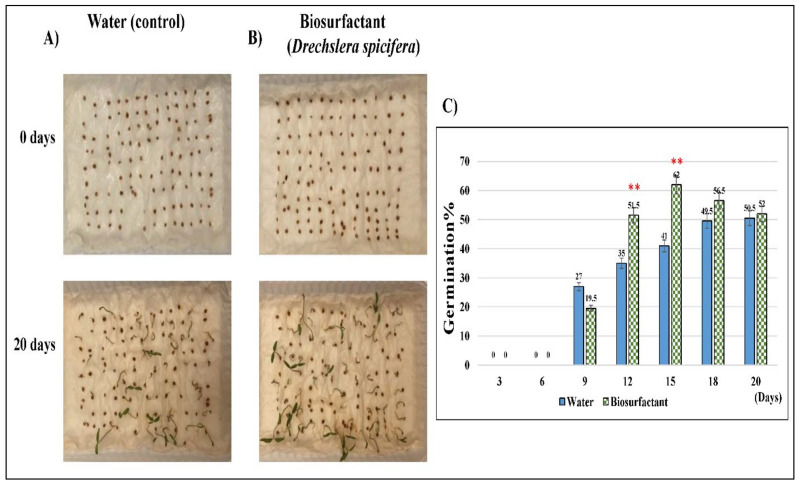
Germination assay of *D. spicifera.* Assessment of toxicity of bio-surfactants produced by *D. spicifera* on the germination of 100 tomato seed for 20 days. (**A**) Untreated seeds (water as control), (**B**) seeds treated with CFS of *D. spicifera*, (**C**) percentage of germination for 20 days. ** indicates significant *p* < 0.01.

**Table 1 molecules-27-06450-t001:** The ability of *D. spicifera* to grow and tolerate different types of petroleum hydrocarbons. The growth rate (cm/day) and the DIR percentages were calculated at 1% concentration of the tested hydrocarbon.

Concentration	Non-Treated (cm/day)	Crude Oil (cm/day)	Used Oil (cm/day)	Diesel (cm/day)	Kerosene (cm/day)	Mixed Oil (cm/day)
1%	3.20	2.24	2.80	1.81	2.00	1.60
5%	3.20	1.15	1.26	1.09	1.05	0.91
10%	3.20	0.55	0.78	0.70	0.62	0.53
DIR (%)	100	70.01	87.50	56.06	62.58	50

**Table 2 molecules-27-06450-t002:** The DCPIP colorimetric analysis, expressed in a.u. at 450 nm, of *D. spicifera* treated with different hydrocarbons.

Incubation Period (Days)	0	3	6	9	12	15
Control	0.000	0.000	0.000	0.000	0.000	0.000
Crude oil	0.000	0.171	0.367	0.486	0.539	0.736
Used oil	0.002	0.272	0.342	0.407	0.454	0.514
Diesel	0.001	0.060	0.109	0.256	0.317	0.401
Kerosene	0.001	0.085	0.177	0.241	0.338	0.405
Mixed oil	0.001	0.095	0.153	0.212	0.301	0.413

**Table 3 molecules-27-06450-t003:** Drop-collapse test to liquid contains surfactants from different strain of *D. spicifera*.

Hydrocarbon Source						
−C	+C	Crude Oil	Used Oil	Diesel	Kerosene	Mixed Oil
−	+++	+	++	+	+	++

−C: negative control (culture broth), +C: positive control (Triton X-100).

## Data Availability

All the data presented in this study are available within the current article. All statistical analysis results and raw data are available on request from the corresponding author.

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
