# Peer review of "Biodegradation of Petroleum Hydrocarbons by Drechsleraspicifera Isolated from Contaminated Soil in Riyadh, Saudi Arabia"

_molecules, 2022, doi:10.3390/molecules27196450_

Round 1

Reviewer 1 Report

1. Introduction should be more descriptive

2. The authors should provide clear figures.

3. Conclusion should be improved.

4. Only microscopic identification of a species is not valid.

5. In table 1 what is the unit of data presented 

Author Response

Reviewer 1 comments and Response

  1. The introduction should be more descriptive

Response: We agree with the reviewer's comment. We have updated the introduction section in the manuscript with the required changes.

  1. The authors should provide clear figures.

Response: According to the instructions of the journal “Figures and Schemes must be provided during submission in a single zip archive and at a sufficiently high resolution (minimum 1000 pixels width/height, or a resolution of 300 dpi or higher). In our manuscript, we used all figures at the resolution of 300 DPI and dimensions of more than 1000 pixels. We suggest the reviewer review the single figures attached with the manuscript, and not only those inserted in the main text. We have updated the labels in figures 2 and 3 to clarify the uncleared information.

  1. The conclusion should be improved.

Response: We agree with the reviewer's comment. We have updated the conclusion section in the manuscript with the required changes.

  1. Only microscopic identification of a species is not valid.

Response: In the current study, the microbial identification of Drecheslera spicifera was carried out by senior microbiology professors (Prof. Fatimah Al-Otibi and prof. Najat Marraiki) according to the microscopic characteristics of the segmentation of the spores and conidia/ Conidiophores structures using the guidelines of the Pictorial atlas of soil and seed fungi of Watanabe (2010). A similar study used the same protocol in which the sporulated fungi were seeded on Potato Dextrose Agar (PDA) plate and only microscopical identification was used [16]. Little modifications were applied in section 4.1.

  1. Abdelwehab, S. A.; EL-Nagerabi, A. A. F.; Elshafie, A. E. Mycobiota Associated with Imported Seeds of Vegetable Crops in Sudan. Open Mycol. J. 2014, 8, 156-173. https://doi.org/10.2174/1874437001408010156
  2. In table 1 what is the unit of data presented?

Response: We agree with the reviewer's comment. We have updated Table 1 in the manuscript with the required changes.

Reviewer 2 Report

The manuscript of Al-Zahrani and coworkers is intended to be an interesting contribution to the field of petroleum (product) contamination remediation. Actually, the present manuscript contains information on the surface-active compound producing, and petroleum (product) degradative ability of a Drechslera (nomenclaturally rather Bipolaris) species. Unfortunately, the only novelty of the manuscript seems to be the use of a filamentous fungus in the studies, in spite of the commonly applied bacteria. There are no methodological or other novelties. Moreover, the applied methods, and the way of application seems to be not adequate, or the results cannot be interpreted as they are. The description of the materials and methods is partly missing. However, let us investigate the manuscript in detail.

The title of the manuscript is not precise enough unfortunately. The results are true for the adequate, investigated strains of the species. Moreover, the name of the species is written erroneously, and there are even nomenclatural problems too.

What concerns the abstract, we have to mention the high number of typists errors (otherwise this is true for the whole manuscript), moreover it is prone to grammatical errors. It is merely a short summary of the results, contains no conclusion from the discussion.

The introduction is blurred, and incomplete. Unfortunately, the existing information on fungal remediation of hydrocarbon pollutants is not summarized; nothing on the possibilities and limitations of their application is mentioned.

Concerning the results. The first statement (chapter 2.1.) describes that the identification of the isolate(s) was merely based on microscopy. Unfortunately, nowadays it is not enough. Molecular analysis or polyphasic identification is the minimum requirement.

Concerning the chapter 2.2. The results of the growth analysis are based on old-fashioned, questionable or even problematic methods. Moreover, the composition of the different tested substrates („oils”) is not described. There are no analytical (e.g. GC) results given, and we do not really know, what the metabolic substrate of the fungus was. There are even SEM images in the manuscript, where arrows highlight something. We do not know what we see. The highlighted points are not described at all. The results of the two growth investigations contradict each other, and in the case of the broth medium, the control is missing.

In chapter 2.3, in most cases the cause and the effect (causality relations) is blurred. In the chapter describing the results of the biosurfactant tests there are again contradictory results, moreover no one knows, why the results of the drop collapse test are not described in this chapter. The results of the tomato seed germination assay are again controversial.

The discussion is blurred, deficient, and mostly erroneous.

The description of the materials and methods is firstly deficient, secondly erroneous. The method of the strain isolation fails; the substrate oils are not described and characterized. How were strains (or only one strain) selected, etc. As one example only if „oil” is the substrate for the „degradation of hydrocarbons as sole source of carbon”, because of the lack of oil characterization one will not know the exact substrate. Might be even that not hydrocarbons were utilized. The description of the hydrocarbon tolerance test is strange and deficient (e.g. the method of inoculation is questionable). Unfortunately, the same is true for the other methodologies too.

Author Response

Response to Reviewer 2 comments

The manuscript of Al-Zahrani and coworkers is intended to be an interesting contribution to the field of petroleum (product) contamination remediation. Actually, the present manuscript contains information on the surface-active compound producing, and petroleum (product) degradative ability of a Drechslera (nomenclatural rather than Bipolaris) species. Unfortunately, the only novelty of the manuscript seems to be the use of a filamentous fungus in the studies, in spite of the commonly applied bacteria. There are no methodological or other novelties. Moreover, the applied methods and the way of application seem to be not adequate, or the results cannot be interpreted as they are. The description of the materials and methods is partly missing. However, let us investigate the manuscript in detail.

Response: In the current study, and for the first time, we evaluated the bio-degradation capabilities of fungus D. spicifera isolated from contaminated soil samples in crude oil reservoirs in Riyadh, Saudi Arabia. After a deep search in the published literature, we found limited studies about the bioremediation abilities of Drechslera sp. against different hydrocarbons. Besides, no studies were conducted in Saudi Arabia, which is one of the most producers of petroleum oils and products. The methodology section at the end of the manuscript is enough in our study as our main goal was to examine the ability of D. spicifera to employ different hydrocarbons as a distinct carbon source which has proven its biodegradation capabilities such as the effect on oil-emulsification, Drop collapse and DCPIP results. Besides, the effect of the isolated biosurfactants on the germination of tomato seeds proves its safe application in different ecosystems. Few changes were made in the introduction section to clarify this point.

The title of the manuscript is not precise enough, unfortunately. The results are true for the adequate, investigated strains of the species. Moreover, the name of the species is written erroneously, and there are even nomenclatural problems too.

Response: We agree with the reviewer's comment. We checked the spelling of Drecheslera spicifera through the manuscript. Changes were done to the title, abstract, and keywords.

What concerns the abstract, we have to mention the high number of typists errors (otherwise this is true for the whole manuscript), moreover, it is prone to grammatical errors. It is merely a short summary of the results and contains no conclusion from the discussion.

Response: We agree with the reviewer's comment. However, according to the journal instructions, the abstract should be a total of about 200 words maximum. We have double-checked the abstract for any possible typists or grammatical errors and corrected them.

The introduction is blurred, and incomplete. Unfortunately, the existing information on fungal remediation of hydrocarbon pollutants is not summarized; nothing on the possibilities and limitations of their application is mentioned.

Response: We agree with the reviewer's comment. We have updated the introduction section in the manuscript with the required changes.

Concerning the results. The first statement (chapter 2.1.) describes that the identification of the isolate(s) was merely based on microscopy. Unfortunately, nowadays it is not enough. Molecular analysis of polyphasic identification is the minimum requirement.

Response: In the current study, the microbial identification of Drecheslera spicifera was carried out by senior microbiology professors (Prof. Fatimah Al-Otibi and prof. Najat Marraiki) according to the microscopic characteristics of the segmentation of the spores and conidia/ Conidiophores structures using the guidelines of the Pictorial atlas of soil and seed fungi of Watanabe (2010). A similar study used the same protocol in which the sporulated fungi were seeded on Potato Dextrose Agar (PDA) plate and only microscopical identification was used [16]. A few modifications were applied in section 4.1.

  1. Abdelwehab, S. A.; EL-Nagerabi, A. A. F.; Elshafie, A. E. Mycobiota Associated with Imported Seeds of Vegetable Crops in Sudan. Open Mycol. J. 2014, 8, 156-173. https://doi.org/10.2174/1874437001408010156

Concerning chapter 2.2. The results of the growth analysis are based on old-fashioned, questionable, or even problematic methods. Moreover, the composition of the different tested substrates („oils”) is not described. There are no analytical (e.g. GC) results given, and we do not really know, what the metabolic substrate of the fungus was. There are even SEM images in the manuscript, where arrows highlight something. We do not know what we see. The highlighted points are not described at all. The results of the two growth investigations contradict each other, and in the case of the broth medium, the control is missing.

Response: We wonder what are the required analysis methods which are not “old-fashioned”. the methodology section had an updated publication as mentioned below

Abdelwehab, S. A.; EL-Nagerabi, A. A. F.; Elshafie, A. E. Mycobiota Associated with Imported Seeds of Vegetable Crops in Sudan. Open Mycol. J. 2014, 8, 156-173. https://doi.org/10.2174/1874437001408010156 

Durairaj, P.; Malla, S.; Nadarajan, S. P.; Lee, P. G.; Jung, E.; Park, H. H.; Kim, B. G.; Yun, H. Fungal cytochrome P450 monooxygenases of Fusarium oxysporum for the synthesis of ω-hydroxy fatty acids in engineered Saccharomyces cerevisiae. Microb. Cell Factories 2015, 14, 1–16. https://doi.org/10.1186/s12934-015-0228-2

Asemoloye, M. D.; Jonathan, S. G.; Ahmad, R. Degradation of 2, 2-Dichlorovinyl dimethyl phosphate (dichlorvos) through the rhizosphere interaction between Panicum maximum Jacq and some selected fungi. Chemosphere 2019, 221, 403-411. https://doi.org/10.1016/j.chemosphere.2019.01.058

Al-Hawash, A. B.; Alkooranee, J. T.; Abood, H. A.; Zhang, J.; Sun, J.; Zhang, X.; Ma, F. Isolation and characterization of two crude oil-degrading fungi strains from Rumaila oil field, Iraq. Biotechnol. Rep. 2018, 17, 104–109. https://doi.org/10.1016/j.btre.2017.12.006 

Al-Otibi, F.; Alkhudhair, S. K.; Alharbi, R. I.; Al-Askar, A. A.; Aljowaie, R. M.; Al-Shehri, S. The Antimicrobial Activities of Silver Nanoparticles from Aqueous Extract of Grape Seeds against Pathogenic Bacteria and Fungi. Molecules 2021, 26, 6081. https://doi.org/10.3390/molecules26196081

Shah, M. U.; Sivapragasam, M.; Moniruzzaman, M.; Yusup, S. B. A comparison of recovery methods of rhamnolipids produced by pseudomonas aeruginosa. Procedia Eng. 2016, 148, 494-500. https://doi.org/10.1016/j.proeng.2016.06.538

Pele, M. A.; Ribeaux, D. R.; Vieira, E. R.; Souza, A. F.; Luna, M. A. C.; Rodríguez, D. M.; Andrade, R. F. S.; Alviano, D. S.; Alviano, C. S.; Barreto-Berger, E.; Santiago, A. L. C. M. A.; Campos-Takaki, G. M. Conversion of renewable substrates for biosurfactant production by Rhizopus arrhizus UCP 1607 and enhancing the removal of diesel oil from marine soil. Electron. J. Biotechnol. 2019, 38, 40–48. https://doi.org/10.1016/j.ejbt.2018.12.003

All of these studies are in the last 6 years, so we thought it’s new, or at least our analysis methods are not “old-fashioned. We have updated the labels in figures 2 and 3 to clarify the uncleared information. The composition of the different tested substrates, and of the metabolic substrate of the fungus is one of the future studies that will be carried out to examine the possible pathways used by this fungus in the bioremediation of the petroleum hydrocarbon as was mentioned in the conclusion section.

In chapter 2.3, in most cases, the cause and the effect (causality relations) is blurred. In the chapter describing the results of the biosurfactant tests there are again contradictory results, moreover, no one knows, why the results of the drop collapse test are not described in this chapter. The results of the tomato seed germination assay are again controversial.

Response: we agree with the reviewer's comment. In chapter 2.3. we tend to discuss how D. spicifera biosurfactants induced hydrocarbons bioremediation. We used the DCPIP results (Supplementary fig. 1), emulsification assay, and drop collapse assay to show the effect of biosurfactants on the physical properties of tested oils. Germination assay was to highlight the safe application of these biosurfactants that didn’t affect the lives and processes of these plants. We have updated the title of Section 2.3. for more clarification. We have updated section 2.5., as well.

The discussion is blurred, deficient, and mostly erroneous.

Response: as we mentioned in the above comment, after a deep search of the published literature, we found limited studies about the bioremediation abilities of Drechslera sp. against different hydrocarbons. Besides, no studies were conducted in Saudi Arabia, which is one of the most producers of petroleum oils and products. So, according to the limited number of references allowed by the journal, we have added the most relevant studies that serve our hypothesis. However, we welcome any suggestions from the reviewer. We have double-checked the discussion for any possible typists or grammatical errors and corrected them.

The description of the materials and methods is firstly deficient, and secondly erroneous. The method of strain isolation fails; the substrate oils are not described and characterized. How were strains (or only one strain) selected, etc? As one example only if „oil” is the substrate for the „degradation of hydrocarbons as the sole source of carbon”, because of the lack of oil characterization one will not know the exact substrate. Might be even that no hydrocarbons were utilized. The description of the hydrocarbon tolerance test is strange and deficient (e.g. the method of inoculation is questionable). Unfortunately, the same is true for the other methodologies too.

Response: please review the above comments.

Reviewer 3 Report

This study is evaluation the the oil-biodegradation ability of the fungus Drecheslera spicifera against Kerosene, Crude, Diesel, Used, and mixed oils. Biodegradation characteristics were evaluated by the following tests: hydrocarbon tolerance, SEM, DCPIP, Drop-Collapse, emulsification activity, biosurfactants recovery, and germination assays. The study showed promising results compared to the literature specially in enhancing the germination of tomato seeds, which I think needs to be explored further.

I have few comments that may be considered before publication:

- is the spelling of "Dracheslera Spicifera" correct? according to NCBI search it is spelled with an "e" to be "Drechslera". Please check.

- The quality of all figures in the manuscript need to be improved. The lables are not easy to read.

- In introduction section, add a brief discussion on the current practice followed to treat the Petroleum hydrocarbons and what are the drawbacks of this current method? and why do you think introducing these biosurfactants would be better?

- section 2.4, check spelling of "weighyt"

- section 4.5, "MSM culture media of was mixed" delete "of"

- The finding of enhancing the seed’s germination needs to be further explained.

Author Response

Reviewer 3 comments and Response

This study is evaluating the oil-biodegradation ability of the fungus Drecheslera spicifera against Kerosene, Crude, Diesel, Used, and mixed oils. Biodegradation characteristics were evaluated by the following tests: hydrocarbon tolerance, SEM, DCPIP, Drop-Collapse, emulsification activity, biosurfactants recovery, and germination assays. The study showed promising results compared to the literature especially in enhancing the germination of tomato seeds, which I think needs to be explored further. I have a few comments that may be considered before publication:

Response: Thanks for the reviewer comment.

- is the spelling of "Dracheslera Spicifera" correct? according to NCBI search, it is spelled with an "e" to be "Drechslera". Please check.

Response: We agree with reviewer comment. We checked the spelling of Drecheslera spicifera through the manuscript. Changes were done to the title, abstract, and keywords.

- The quality of all figures in the manuscript needs to be improved. The labels are not easy to read.

Response: According to the instructions of the journal “Figures and Schemes must be provided during submission in a single zip archive and at a sufficiently high resolution (minimum 1000 pixels width/height, or a resolution of 300 dpi or higher). In our manuscript, we used all figures at the resolution of 300 DPI and dimensions of more than 1000 pixels. We suggest the reviewer review the single figures attached with the manuscript, and not only those inserted in the main text. We have updated the labels in figures 2 and 3 to clarify the uncleared information.

- In the introduction section, add a brief discussion on the current practice followed to treat Petroleum hydrocarbons and what are the drawbacks of this current method. and why do you think introducing these biosurfactants would be better?

Response: We agree with reviewer comment. We have updated the introduction section in the manuscript with the required changes.

- In section 2.4, check the spelling of "weighyt"

Response: We agree with reviewer comment. We have updated the manuscript with the required changes. Changes were done in the sub-heading title in section 2.4. as advised.

- section 4.5, "MSM culture media of was mixed" delete "of"

Response: We agree with reviewer comment. We have updated the manuscript with the required changes. Changes were done in section 4.5. as advised.

- The finding of enhancing the seed’s germination needs to be further explained.

Response: We agree with the reviewer's comment. We have updated the manuscript with the required changes. Changes were done in section 2.5. as advised.

Round 2

Reviewer 2 Report

Unfortunately, the revision was only superficial. No substantive changes were made. I.e. the species name of the fungus is still misspelled; the identification is not adequate; the description of the methodology is incomplete and in part wrong; the results are based on wrong methods; the results give practically no novelty, and are based on one single strain; the discussion lacks the diagnosis of the fungus, the precise applicability. I have to state, that something then this could even not be defended as an masters' thesis. Thus the publication has to be rejected. I am sorry, but this is the case. It is not only not enough, but wrong.

Author Response

Response to Reviewer 2 comments

  1. Unfortunately, the revision was only superficial. No substantive changes were made. I.e., the species name of the fungus is still misspelled;

Response: we agree with the reviewer comment we have found few mistakes in lines

  1. the identification is not adequate;

Response: Again, we disagree with the reviewer comment. The microbial identification of Drechslera spicifera was carried out by senior microbiology professors (Prof. Fatimah Al-Otibi and prof. Najat Marraiki) according to the microscopic characteristics of the segmentation of the spores and conidia/ Conidiophores structures using the guidelines of the Pictorial atlas of soil and seed fungi of Watanabe (2010). This is common and reliable; besides, it was used as a single identification method that was published before. For instance, the recent publication in IOP Conference Series: Earth and Environmental Science:

Hanum S, Fitri L, Lisa O et al. Inventory of fungi from termite nests at Gunung Leuser National Park, northern Sumatra. IOP Conference Series: Earth and Environmental Science. 2021;667(1):012088. doi:10.1088/1755-1315/667/1/012088

They used the microscopic method in the identification of the isolated fungi.

Furthermore, another study from Italy used the same identification method for different strains isolated form wooden artworks and canvases.

Sabatini L, Sisti M, Campana R. Evaluation of fungal community involved in the bioderioration process of wooden artworks and canvases in Montefeltro area (Marche, Italy). Microbiol Res. 2018 Mar;207:203-210. doi: 10.1016/j.micres.2017.12.003. Epub 2017 Dec 8. PMID: 29458856.

So, actually some recent publications used only the microscopic identification, besides, we add the SEM imaging results of D. spicifera, in which the untreated control is shown in figures3A and 3B. To clarify this point, we have replaced some sentences in the discussion with the above references (Highlighted).

  1. the description of the methodology is incomplete and in part wrong;

Response: Again, we disagree with the reviewer comment. We have cited all of the used methods with references. We add the details of the materials, instruments, conditions. calculations in each technique. We think, it’s not required to add the detailed protocol in each method. There are no changes have been made in the current version, otherwise, the reviewer has to mention exactly, what is required to be corrected.

  1. the results are based on wrong methods;

Response: Again, we disagree with the reviewer comment. Please review the above responses.  There are no changes have been made in the current version, otherwise, the reviewer has to mention, exactly, what is required to be corrected.

  1. the results give practically no novelty, and are based on one single strain;

Response: We disagree with the reviewer comment. Many previous publications studied the biodegradation abilities of a single strain. Examples:

Wang XB, Chi CQ, Nie Y, Tang YQ, Tan Y, Wu G, Wu XL. Degradation of petroleum hydrocarbons (C6-C40) and crude oil by a novel Dietzia strain. Bioresour Technol. 2011 Sep;102(17):7755-61. doi: 10.1016/j.biortech.2011.06.009. Epub 2011 Jun 12. PMID: 21715162.

Zhang L, Zhang C, Cheng Z, Yao Y, Chen J. Biodegradation of benzene, toluene, ethylbenzene, and o-xylene by the bacterium Mycobacterium cosmeticum byf-4. Chemosphere. 2013 Jan;90(4):1340-7. doi: 10.1016/j.chemosphere.2012.06.043. Epub 2012 Sep 7. PMID: 22960059.

Yakimov MM, Giuliano L, Gentile G, Crisafi E, Chernikova TN, Abraham WR, Lünsdorf H, Timmis KN, Golyshin PN. Oleispira antarctica gen. nov., sp. nov., a novel hydrocarbonoclastic marine bacterium isolated from Antarctic coastal sea water. Int J Syst Evol Microbiol. 2003 May;53(Pt 3):779-785. doi: 10.1099/ijs.0.02366-0. PMID: 12807200.

Varjani SJ, Upasani VN. Biodegradation of petroleum hydrocarbons by oleophilic strain of Pseudomonas aeruginosa NCIM 5514. Bioresour Technol. 2016 Dec;222:195-201. doi: 10.1016/j.biortech.2016.10.006. Epub 2016 Oct 3. PMID: 27718402.

  1. the discussion lacks the diagnosis of the fungus, the precise applicability.

Response: we agree with the reviewer comment. To clarify this point, we have replaced some sentences in the discussion with the above references in response number 2 (Highlighted).

I have to state, that something then this could even not be defended as an masters' thesis. Thus, the publication has to be rejected. I am sorry, but this is the case. It is not only not enough, but wrong.

Response: this is the reviewer's view that we respect, however, we refuse entirely it. The current paper is a novel study about the biodegradation ability of Drechslera spicifera. That hasn’t been published before (If you find any please attach it). The manuscript is full of many details and methods to improve our hypothesis. Again, we disagree with the reviewer's comment. Please review the above responses.  There are no changes have been made in the current version, otherwise, the reviewer has to mention, exactly, what is required to be corrected.
